# Is School Gardening Combined with Physical Activity Intervention Effective for Improving Childhood Obesity? A Systematic Review and Meta-Analysis

**DOI:** 10.3390/nu13082605

**Published:** 2021-07-28

**Authors:** Yufei Qi, Sareena Hanim Hamzah, Erya Gu, Haonan Wang, Yue Xi, Minghui Sun, Siyu Rong, Qian Lin

**Affiliations:** 1Centre for Sport and Exercise Sciences, Universiti Malaya, Kuala Lumpur 50603, Malaysia; yufeiqi@csu.edu.cn (Y.Q.); sareena@um.edu.my (S.H.H.); 2Department of Physical Education and Research, Central South University, 932 Lushan South Rd., Changsha 410083, China; haonanwangcsu@csu.edu.cn; 3School of Foreign Languages, Central South University, 932 Lushan South Rd., Changsha 410083, China; 8106180507@csu.edu.cn; 4Department of Nutrition Science and Food Hygiene, Xiangya School of Public Health, Central South University, 110 Xiangya Rd., Changsha 410078, China; xiyue0404@csu.edu.cn (Y.X.); sun.1234@csu.edu.cn (M.S.); 5Graduate School, Wuhan Sports University, Wuhan 430079, China; s2024654@siswa.um.edu.my

**Keywords:** physical activity, school gardening activity, childhood obesity, meta-analysis, systematic review

## Abstract

School gardening activities (SGA) combined with physical activities (PA) may improve childhood dietary intake and prevent overweight and obesity. This study aims to evaluate the effect of SGA combined with PA on children’s dietary intake and anthropometric outcomes. We searched studies containing randomized controlled trials up to January 2021 in Web of Science, PubMed, Cochrane Library, and the EBSCO database on this topic for children aged 7 to 12 years. Fourteen studies met the requirements for meta-analysis (*n* = 9187). We found that SGA has no obvious effect on improving children’s BMI (WMD = −0.49; *p* = 0.085; I^2^ = 86.3%), BMI z-score (WMD = −0.12; *p* = 0.235; I^2^ = 63.0%), and WC (WMD = −0.98; *p* = 0.05; I^2^ = 72.9%). SGA can effectively improve children’s FVs (WMD = 0.59, *p* = 0.003, I^2^ = 95.3%). SGA combined with PA can significantly increase children’s FVs but cannot greatly improve weight status. Although more studies on this topic are needed to prove the effectiveness of this method, the results of our review show that both SGA and SGA combined with PA has a modest but positive impact of reducing BMI and WC outcomes but can significantly increase children’s FVs.

## 1. Introduction

Childhood obesity is a public health priority and its prevalence and economic burden have been steadily increasing worldwide [1,2]. In the past three decades, the detection rate of childhood obesity in the United States has tripled, and the rate of adolescent obesity detection has increased by four times [3,4,5]. Related reports showed that in the past 20 years, the body mass index (BMI) of children and adolescents aged 6 to 17 in China has increased by 11.1% for overweight and 7.9% for obesity [6]. Childhood obesity is strongly associated with adult obesity [7,8,9]. It not only has an impact on children’s physical and mental health, as well as their social abilities, but it is also the primary risk factor for metabolic syndrome, cardiovascular- and cerebrovascular-related chronic diseases [10,11,12,13], and death in adulthood [14].

Public institutions have developed and implemented several strategies to prevent obesity in children, including school gardening activities (SGA) and physical activities (PA). SGA, known as a “learning laboratory”, is a way based on Social Cognitive Theory (SCT) to teach students’ skills through fun hands-on activities, selecting the school as the intervention site and using gardening as a key component is a promising approach to addressing healthy eating and student’s weight status [15,16,17]. In school gardens where students grow edible produce, they generally learn science and nutrition concepts relevant to growing food while they work in the garden, which enhances students’ positive modeling of fruits and vegetables and increases their preference for these foods as well as their fruit and vegetable intakes (FVs) [18,19]. According to Molitor and Doerr [20], children who learn healthy eating habits at a young age are more likely to maintain these habits throughout their life. Children at a young age must develop healthy eating habits for their future health [21]. Some scholars have used SGA to successfully improve children’s FVs and their intake preferences [22,23,24], but the opinions on the effects of SGA on weight are divided [25,26,27,28]. Scientific PA is an effective way to reduce BMI, body fat, and serum Chemerin in obese children, and also controls their blood pressure [29,30]. Short-term or long-term moderate-to-high intensity physical activity (MVPA) has been related to increased blood flow to the brain and neurotransmitter levels, as well as enhanced attention, motor skills, physical fitness, executive function, and social skills, and improved mental health in children [31,32,33,34]. Extracurricular PA is also considered closely during the study process, since the short time of school physical education may be occupied by other key courses [35,36]. However, the research shows that extracurricular PA has mixed effects on improving children’s obesity [37,38,39]. Considering that each SGA and PA intervention was proven to be effective, theoretically, combining SGA with PA might improve children’s eating habits and increase the amount of PA to reduce obesity more effectively.

BMI, body mass index z-score (BMI z-score), and WC are commonly used to measure anthropometric characteristics and stratify the risk for overweight and obesity in adults and children [40,41]. BMI and WC are easily obtainable and have proved to be strong predictors of metabolic syndrome, type 2 diabetes, and cardiovascular disease in adults and children [42]. Many scholars of the United States and other developed countries have used SGA to explore its impact on children’s BMI and other anthropometric outcomes [43,44,45]. Therefore, we systematically reviewed and meta-analyzed the effects of SGA on anthropometric outcomes of school-age children (7–12 years old), gathering experimental information on BMI and WC changes in SGA and other interventions. Additionally, we undertook an SR on the impact of SGA and PA on the motivation and preference of fruits and vegetables and anthropometric outcomes of school-age children (7–12 years old). This work can provide clinicians, teachers, and policymakers with robust evidence on the efficacy of SGA combined with PA as a comprehensive intervention model to improve children’s obesity, providing an important countermeasure to deal with childhood obesity.

## 2. Materials and Methods

### 2.1. Databases and Search Criteria

We performed a systematic search and review according to PRISMA guidelines [46]. We searched Web of Science, PubMed, Cochrane Library, and EBSCO databases to identify relevant studies as for January 2021. Additionally, the specifically designed search strings for this purpose were “obesity”, “children”, “school gardening”, “physical activities”, and “fruits and vegetables”. W.H.N. and R.S.Y. completed the research process individually and in duplicate. Meeting and consultations with another author (Q.Y.F) solved any disagreements in these regards. Our research was registered in PROSPERO (CRD42021232858) on 28 March 2021.

### 2.2. Study Selection

Inclusion criteria were as follows: (1) children attending school, aged 7–12 years; (2) SGA and PA for obesity prevention; (3) only SGA or SGA combined with PA for intervention groups, only nutritional education or no intervention for control groups, and post-intervention outcomes including FVs, BMI, BMI z-score, and WC; and (4) randomized controlled trials (RCTs).

Exclusion criteria were as follows: (1) research on gardening activity outside school, (2) studies in which FVs outcomes were not recorded accurately by food intake diaries or Block Kids food filters, (3) only FVs as an outcome, and (4) repeated study.

### 2.3. Data Selection and Extraction

Two authors (W.H.N. and R.S.Y.) extracted data from the included studies for analysis, resolved any disagreement through discussion and reaching a consensus, or requested an opinion of a third author (Q.Y.F). The extracted outcomes included author, publication year, number of study participants, mean age, intervention measures, outcomes, study duration, etc.

### 2.4. Study Quality Assessment

Two authors (X.Y. and S.M.H.) evaluated the quality of the included studies using the Cochrane Collaboration Risk of Bias (ROB2.0) for assessing the risk of bias recommended. The Cochrane Tool allowed for analyzing the following groups: random sequence generation, allocation concealment, blinding of participants and personnel, blinding of outcome assessment, incomplete outcome data, selective reporting, and other (analysis for intention to treat and compliance). For each bias group, it was possible to assign a value of “high”, “low”, or “unknown” risk of bias when it was not specified if a specific bias was present or not. Any disagreements in the evaluation process were resolved by the third author (Q.Y.F.).

### 2.5. Statistical Analysis

We applied Stata 16.0 software to analyze the heterogeneity of studies and calculate the I^2^ statistic. If the I^2^ statistic was greater than 50%, we utilized a random-effects model, otherwise a fixed-effects model was used. Outcome measures were merged with the effect value Z, the weighted mean difference (W.M.D.), and the confidence interval value (95% CI). Results of the meta-analysis were visualized using forest plots, while the publication bias was assessed using funnel plot and Egger’s test. We also conducted a sensitivity analysis of the combined results of I^2^ studies to find the sources of heterogeneity and corresponding reasons.

## 3. Results

### 3.1. Search Findings

Our search yielded 216 studies, among which 202 studies were retrieved by database browsing. We included 14 studies that met the eligibility criteria, 12 of which were appropriate for meta-analysis and 2 for review. Figure 1 shows the flowchart with the study selection process.

### 3.2. Characteristics of the Included Studies

All studies included designed SGA for primary school-age children. The duration of SGA ranged from 10 to 52 weeks. The total sample sizes ranged from 102 to 3153 children, among which most of the sample size ranged from 100 to 400 children. SGA provided opportunities for children to plant, water, weed, harvest, and taste various fruits and vegetables. Other interventions integrated activities, such as nutritional education, cooking activities, and participating in sports. The interventions included in the 14 studies can be divided into two types: one based on the SGA or GA only (*n* = 12), and one with an intervention including both SGA and PA (*n* = 2). Detailed studies’ information are shown in Table 1.

### 3.3. Quality Assessment

The results of Cochrane’s risk of bias assessment (Figure 2 and Figure 3) showed that for most studies, the overall risk of bias was low. Among them, all of the studies (100%) described the generation of random sequences in detail. In the categories “Timing of identification or recruitment of participants” and “Deviations from the intended interventions”, 22 (79%) trials were assessed as low risk and 6 (21%) trials were assessed as unclear risk, which was related to the lack of accurate information on blinding. The low-risk categories in 64%–86% of the studies were “Selection of the reported result”, “Measurement of the outcome”, and “Missing outcome data”, and the remaining studies were judged to be of unclear risk due to insufficient information on the methods used by researchers to randomly assign participants to groups and in reporting all predefined results. Due to the limited number of studies, it was impossible to explore the existence of publication bias.

### 3.4. FVs

We analyzed the FVs comparison results between the SGA groups and control groups (random model I^2^ = 96.2%). The results showed that FVs in both intervention groups and control groups increased in different degrees, and the difference in added value had statistical significance (WMD = 0.59, Z = 3.01, *p* = 0.003, 95% CI = −0.21~0.98) (Figure 4). The sensitivity analysis outcomes (Appendix A) show that the source of heterogeneity is Davis’ study.

### 3.5. BMI

Figure 5 shows the comparison results of the BMI between SGA groups and control groups (random model I^2^ = 86.3%). The difference in BMI reduction between the intervention groups and control groups showed no statistical significance (WMD = −0.49, Z = 0.84, *p* = 0.404, 95% CI = −1.63–0.65). Sensitivity analysis indicated that Davis’ study is a source of heterogeneity in BMI combined results (Appendix A).

### 3.6. BMI z-Score

The comparison results of the BMI z-score between the SGA groups and control groups is shown in Figure 6 (random model I^2^ = 63.0%). The results turned out to show that there was no statistical significance in the reduction of the BMI z-score between the intervention groups and control groups (WMD = −0.12, Z = 1.72, *p* = 0.085, 95% CI = −0.26–0.02). We carried out a sensitivity analysis (Appendix A) and Gatto’s study is also a source of heterogeneity in the BMI z-score.

### 3.7. WC

We analyzed the comparison results of WC between the SGA intervention groups and control groups (random model I^2^ = 72.9%). The results turned out to show that there was no statistical significance in the reduction of WC between intervention groups and control groups (WMD = −0.98, Z = 1.19, *p* = 0.235, 95% CI = −2.61–0.64) (Figure 7). We carried out sensitivity analysis on all of the six included studies, and the results showed that Gatto’s study was the source of heterogeneity in the WC combined results (Appendix A).

### 3.8. Results Impacted by SGA Combined with PA

A study by Evans et al. [54] set the main outcome as children’s BMI alongside other outcomes, such as vegetable intake motivation, vegetable preference, vegetable consumption, sugar-sweetened beverage consumption, and engagement PA. They divided experimental subjects into three intervals according to their BMI index—normal weight, overweight, and obesity—and evaluated the change of obesity by counting the proportion of people in each interval before and after the intervention. The results showed that there was no significant difference in overweight and obesity intervals between the intervention and control group (*p* = 0.8), and subjects’ vegetable intake, intake motivation, and preference in SGA combined with the PA group and the SGA group were significantly improved compared with the control group (*p* = 0.3). Alexandra et al. [55] used the same intervention methods as Evans’ [54] to test the accuracy of the research results. After 24 weeks of intervention, they found that vegetable preference in the SGA combined with PA group, as well as in the SGA group, was significantly improved compared to the control group (*p* = 0.013 and *p* = 0.001, respectively), especially in regard to vegetable taste (*p* = 0.000 and *p* = 0.000, respectively) and nutritional knowledge (*p* = 0.033, *p* = 0.001 respectively)”. Additionally, compared with the control group, the proportions of overweight and obese subjects in the SGA group (*p* = 0.039) and PA group (*p* = 0.042) were significantly lower, respectively, but there was no significant difference between the SGA combined with PA group and the control group. 

The publication bias was assessed by funnel plot and egger’s test (Appendix A).

## 4. Discussion

This is the first SR and meta-analysis combining SGA with PA to evaluate the changes in the obesity index of children, such as FVs, BMI, BMI z-score, and WC. Since most of the current studies only use an SGA as the main method, more data on the obesity index can be collected. Still, these studies were included in this study for meta-analysis. However, the only two studies using SGA combined with PA as the intervention method have included FVs and other obesity-related outcomes [54,55], which were not unified with those outcomes in the other 12 studies. Thus, they can only be included in this study for SR. Our study shows that SGA and SGA combined with PA can effectively increase children’s FVs and improve their fruit and vegetable knowledge, intake motivation, and intake preference. The above-mentioned results are in accordance with previous studies’ results [15,56,57,58,59] for these outcomes, and highlight this strategy’s importance in cultivating children’s good eating habits and improving their weight.

The outcomes, such as children’s knowledge of fruits and vegetables, intake preference, and intake motivation, could not accurately measure the changes of children’s FVs. Therefore, some researchers use a 24 h food diary (CADET) or Block Kids food filter to measure the changes of FVs in portions [23,25,26,47,53]. Among them, Christian’s [23] and Davis’ [53] studies had the most significant effects and showed an increase in the children’s FVs by about 1.4 and 0.8 portions through 72 weeks and 36 weeks of SGA and other interventions, respectively. While in Morgan’s [26] and Hanbazaza’s [47] studies, after 10 and 72 weeks of SGA, the FVs increased only by 0.2 and 0.4 portions, respectively. The reasons for this difference may be as follows. Christian [23] strengthened the role of parents in SGA. Parents not only participated in part of the curriculum of the SGA, but were also supervising their children to fill in the CADET, which improved the results helpfully. Professional gardening experts were used in Davis’ [53] study as interveners who conducted SGA for children with cooking activities and nutrition education as auxiliary means, which effectively enhanced children’s FVs. The short intervention time of Morgan’s [26] and Ratcliffe’s [25] studies may have affected the outcomes. Although Hanbazaza [47] also conducted a 72-week SGA, the frequency of intervention and parental participation was low, which may be an important reason for limiting an effective improvement of the outcomes. However, the above studies have confirmed that SGA has a positive effect on improving children’s FVs.

The meta-analysis of anthropometric outcomes related to obesity showed that the BMI, BMI z-score, and WC were all reduced to varying degrees after the intervention. A recent study [53] with an expanded sample size and more rigorous statistics found that SGA had no obvious effect on the three outcomes. Therefore, we concluded that overall results indicated that SGA had no significant effect on improving children’s obesity. The World Health Organization (WHO) has analyzed the possibility and scientific validity of using BMI, WC, and the Waist-Hip Ratio to predict chronic diseases [15]. Additionally, the decrease in BMI and WC after intervention in some studies [22,27,48,49,50,52,60] indicated that SGA has a positive effect on the prevention and improvement of some chronic diseases in children, even if the improvement effect is limited. Among them, five studies [22,27,48,50,53] added cooking intervention into SGA, namely cooking demonstrations and having children cook vegetables and fruits planted by themselves. The results showed that compared with the outcomes of those only intervened by SGA, the BMI and WC outcomes of those intervened by both cooking intervention and SGA reduced more significantly. For example, Davis et al. [53] added cooking demonstrations and other activities to the SGA and made them into the LA Sprouts’ course. The results showed that the BMI and WC of children in the LA Sprouts’ course group decreased more significantly than the SGA group or cooking intervention group separately. This coincided with the conclusion that SGA combined with nutrition education could more effectively prevent child obesity. Both proved that combined interventions can achieve more effective effects [16,51], which provides new ideas for improving the health of obese children. The results of the meta-analysis indicated that interventions based only on SGA had no significant impact on children’s BMI and WC. In other words, a single intervention such as gardening or cooking has limited effect on improving childhood obesity, so we should consider combing SGA with other interventions in the future to explore its effectiveness in improving overweight and obesity in children.

As highlighted by Katz et al. [16,61], the 6 h that school-aged children spend at school every day, for more than half the year, constitute a substantial part of their time and their lives. Therefore, it is important to consider school as one of the major drivers and elective settings for children’s education on healthy lifestyles [62,63]. Recently, more studies have shown that physical activities provide many benefits in regulating metabolism and weight loss by increasing energy consumption and improving metabolic status [64,65,66,67,68], so more attention was paid to physical exercise-related courses in the program of preventing childhood obesity. In theory, SGA paired with PA can improve the children’s FVs and their PA, making it more successful in reducing obesity in children.

There are only two studies on SGA combined with PA, but these two studies only measured children’s intake of vegetables without the intake of fruits. This may be because vegetables are relatively easy to grow and harvest. In addition, the two studies only measured the proportion of people in different weight intervals in anthropometrics, which may be related to the difficulty in collecting data of the large sample size selected in the study. Evans et al. [54] conducted a 48-week SGA combined with PA and found that children’s preference for vegetable intake increased significantly, but the BMI outcome showed that the number of overweight and obese children did not significantly decrease. Since the sample size selected by the researcher was relatively large and the samples were distributed in 28 schools, it was difficult to carry out the intervention and balance all conditions. However, this combination of SGA and PA is an innovation. On this basis, Alexandra et al. [55] did their study with the same intervention method. Additionally, the results showed that both the SGA group and SGA combined with the PA group effectively improved children’s vegetable-related outcomes. Both the SGA group and PA group effectively improved childhood obesity. However, the SGA combined with PA had no obvious effect on improving childhood obesity, which may be related to the implementation of the intervention. Because the curriculum executors of the SGA combined with the PA group were not teachers but trainers, the implementation of the intervention may lack scientific validity and rationality. In addition, these two studies both used low-intensity PA. However, many studies have shown that compared to low-intensity PA, MVPA has a significant effect on reducing obese children’s BMI, WC, and other outcomes [69,70,71,72,73]. In future studies, MVPA should be combined with SGA, and the obesity outcomes, such as WC and blood pressure, should be added to more comprehensively detect the accuracy of obesity improvement, which provides a new perspective and ideas for future studies.

This study has the following limitations. First, although the number of studies on SGA has steadily increased in the last 10 years, there were few comprehensive studies on SGA combined with PA, which should be explored in the future. Second, FVs-related outcomes in most studies [23,25,26,47,53] were measured by self-report, which is easily affected by social recognition bias, so they may not accurately represent the changes in dietary intake. Thirdly, there are little data on long-term changes of FVs, so we were unable to determine if the changes of FVs continued over time, and further research is needed. Finally, the main purpose of gardening-based interventions is to improve children’s intake of fruits and vegetables. Therefore, the literature included in this study only analyzed changes in fruit and vegetable intake. In future studies, we can also observe whether unhealthy eating behaviors related to childhood obesity have improved, such as excess intake of sugar-sweetened beverages, desserts, and fried foods.

## 5. Conclusions

The intervention based on SGA can effectively increase children’s FVs and improve their intake motivation, attitude, and preference for vegetables and fruits, but it has no obvious effect on reducing BMI outcomes and WC. The increase in both FVs and PA has obvious effects on the improvement of childhood obesity. In future studies, we should consider integrating various interventions such as SGA, PA, and cooking. Additionally, we can reasonably learn from and innovate the research design of previous scholars and formulate more scientific research plans to explore methods that are more conducive to improving childhood obesity.

## Figures and Tables

**Figure 1 nutrients-13-02605-f001:**
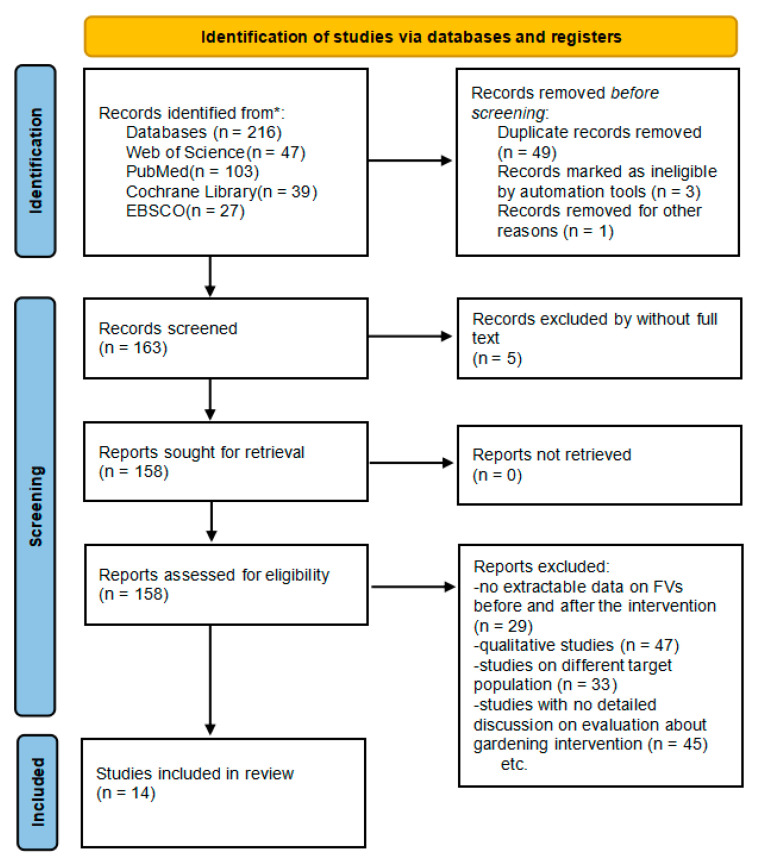
Flow diagram of studies selection. * Consider, if feasible to do so, reporting the number of records identified from each database or register searched (rather than the total number across all databases/registers).

**Figure 2 nutrients-13-02605-f002:**
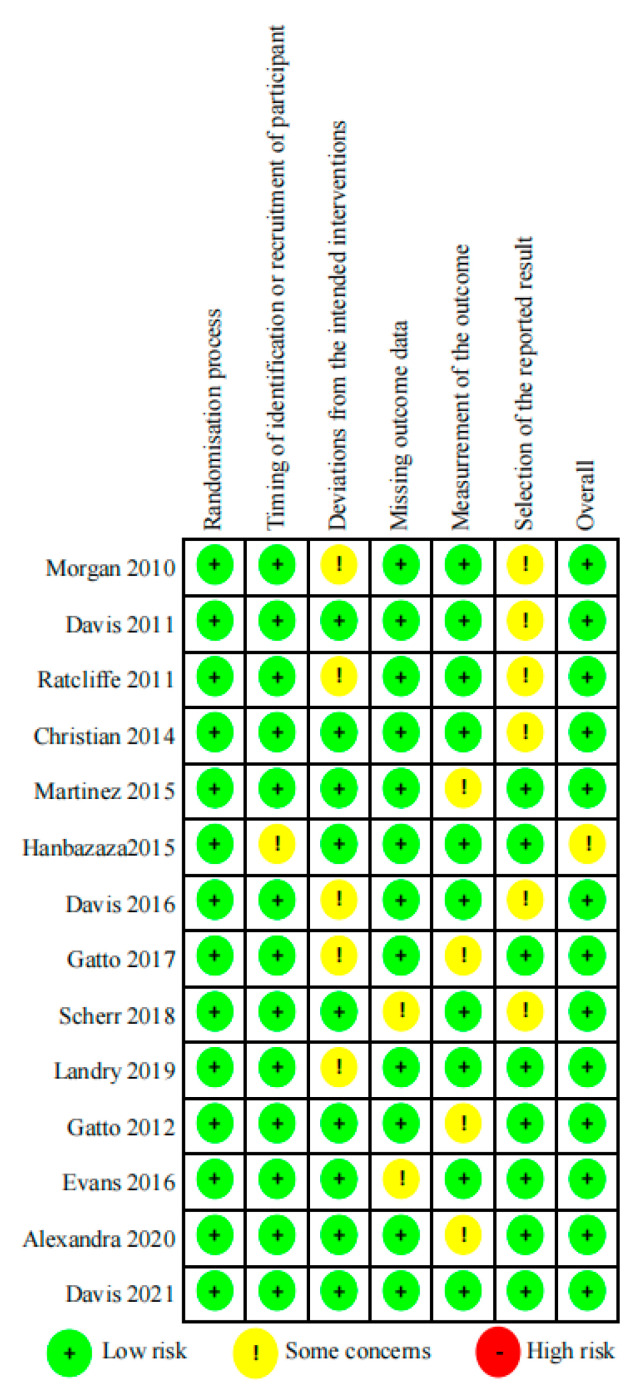
Risk of bias graph per type of bias assessed.

**Figure 3 nutrients-13-02605-f003:**
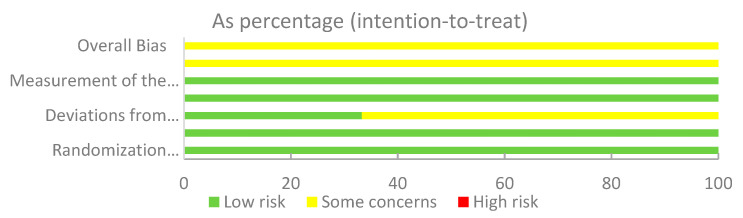
Risk of bias summary for the studies assessed.

**Figure 4 nutrients-13-02605-f004:**
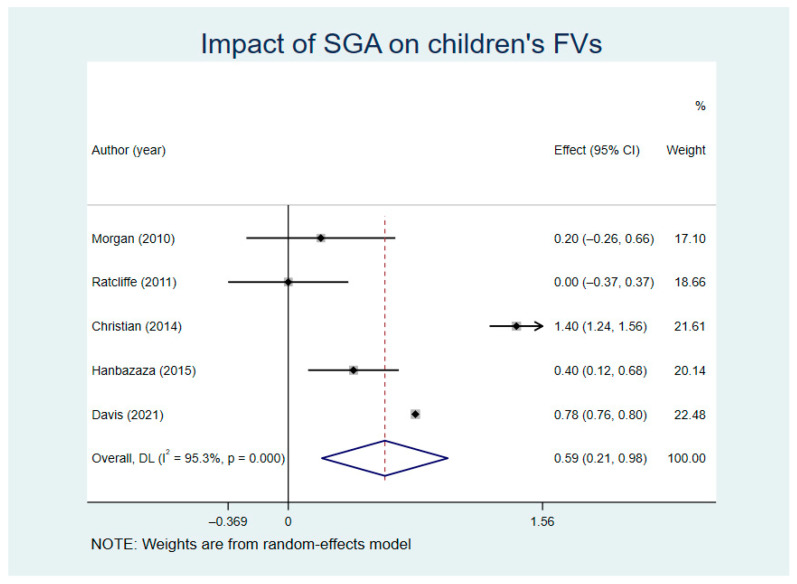
Impact of SGA on children’s fruit and vegetable intakes (FVs).

**Figure 5 nutrients-13-02605-f005:**
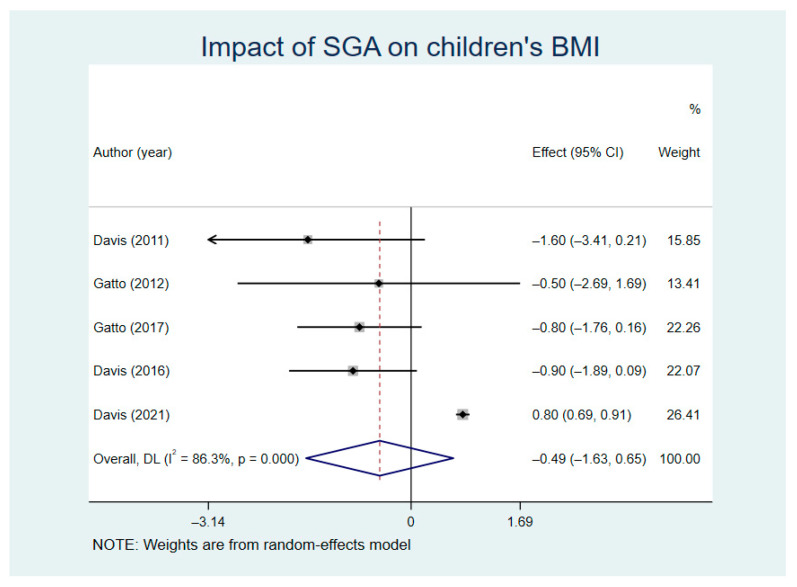
Impact of school gardening activities (SGA) on children’s body mass index (BMI).

**Figure 6 nutrients-13-02605-f006:**
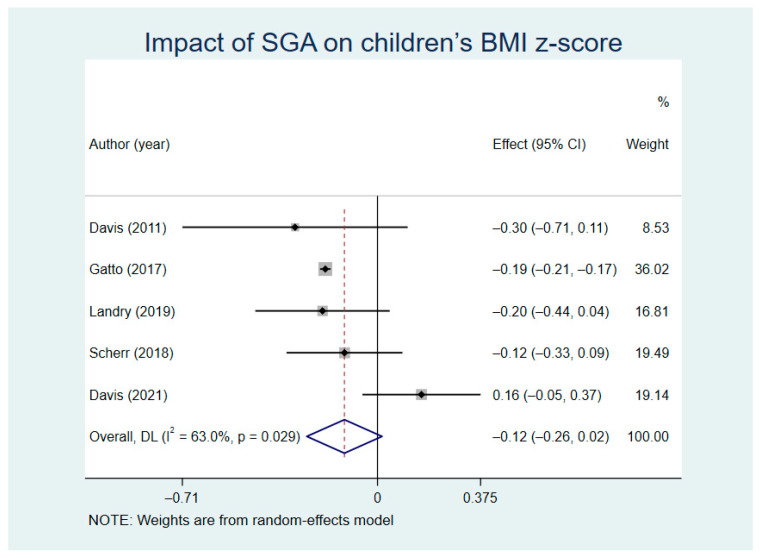
Impact of SGA on children’s body mass index z-score (BMI z-score).

**Figure 7 nutrients-13-02605-f007:**
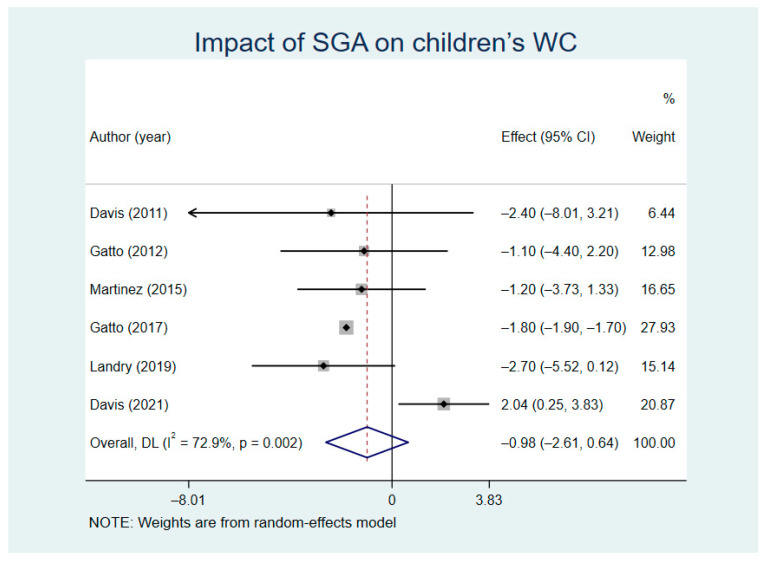
Impact of SGA on children’s waist circumference (WC).

**Table 1 nutrients-13-02605-t001:** Characteristics of the included studies.

First Author	Year	Sample Size	Age	Interventions	Study Duration	Control Group	Outcomes	Results
Davis [22]	2011	114	7–12	Gardening	12 weeks	No intervention	BMIz BMI WC	BMI↓ BMIz↓ WC↓
Christian [23]	2014	641	7–11	Gardening	72 weeks	No intervention	FVs	FVs↑
Ratcliffe [25]	2011	320	12	Gardening	13 weeks	No intervention	FVs	FVs -
Morgan [26]	2010	127	11–12	Gardening	10 weeks	No intervention	FVs	FVs↑
Gatto [27]	2017	319	9–11	Gardening	12 weeks	No intervention	BMI BMIz WC	BMI↓ BMIz↓ WC↓
Hanbazaza [47]	2015	116	7–12	Gardening	72 weeks	No intervention	FVs	FVs↑
Davis [48]	2016	304	9–11	Gardening	12 weeks	No intervention	BMI	BMI↓
Landry [49]	2019	290	9–11	Gardening	12 weeks	No intervention	BMIz WC	BMIz↓ WC↓
Martinez [50]	2015	364	9–11	Gardening	12 weeks	No intervention	WC	WC↓
Scherr [51]	2018	409	9–10	Gardening	48 weeks	No intervention	BMIz	BMIz↓
Gatto [52]	2012	104	9–11	Gardening	12 weeks	No intervention	WC BMI	WC↓ BMI↓
Davis [53]	2021	3153	9–11	Gardening	36 weeks	No intervention	FVs BMIz BMI WC	FVs↑ BMIz↑ BMI↑ WC↑
Evans [54]	2016	1600	7–9	Gardening and Physical Activity	48 weeks	No intervention	others	
Alexandra [55]	2020	1326	9–12	Gardening and Physical Activity	24 weeks	No intervention	others	

## Data Availability

Data are contained within the paper.

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
