# Peer review of "Is School Gardening Combined with Physical Activity Intervention Effective for Improving Childhood Obesity? A Systematic Review and Meta-Analysis"

_nutrients, 2021, doi:10.3390/nu13082605_

Round 1

Reviewer 1 Report

Thank you for the opportunity to review this manuscript. I have the following major and minor comments for the authors to consider.

Specific comments:

  1. As per the journal's guidelines, the abstract should be a total of about 200 words maximum. The abstract should be a single paragraph and should follow the style of structured abstracts, but without headings.
  2. The study title is slightly convoluted. Suggest to make it more concise and succinct.
  3. Please provide the actual p value rather than simply "P>0.05" or "P<0.05". This is neither informative nor useful for readers.
  4. The general health benefits of exercise, be it a short-bout or long-term should be mentioned in the introduction or discussion section. For example, exercise has been linked to increased blood flow to the brain and neurotransmitter levels, enhanced plasticity and better focus, attention and information processing in children (citation: pubmed.ncbi.nlm.nih.gov/28917364).
  5. Please append a copy of the completed PRISMA checklist to the submission or additional files.
  6. When PubMed is used for the search, MESH terms are always recommended to be included. Please also provide the full electronic search strategy used to identify studies, including all search terms and limits for at least one database.
  7. The methods used were not adequately described; exactly who did what to identify, review, assess and resolve disagreements in the identified manuscripts. 
  8. More information about the "Gardening" intervention is required, beyond just generic descriptions such as "opportunities for children to plant, water, weed, harvest and taste various fruits and vegetables."
  9. There was no funnel plot or assessment for potential publication bias.
  10. Did the children's dietary habits also change as a result of their participation in gardening?
  11. Please change "we can’t determine whether" to "we were unable to determine if".

Reviewer 2 Report

An interesting and relevant topic. 

Abstract

Well-written summary

Introduction

The introduction implies it's a SRMA done with studies conducted in China. However, the title and abstract are not reflecting this. Similarly, the methods and results were not confined to China. 

Methods

Please include the initials of the authors who conducted the screening, extraction, and quality assessment. 

Include an assessment of publication bias.

Results

Update Figure 1 to the latest PRISMA flow diagram

It's unclear if the authors have used ROB or ROB2 in their quality assessment. Authors are advised to use the latest risk of bias tool ROB2.0. 

The forest plots appear to be generated using different software eg. Fig 2 vs Fig 3. If it's the case, please disclose this in methods. 

Forest plots of sensitivity analysis can be presented as supplementary docs as it's disrupting the flow of the results writing. 

Discussion & conclusion

No further comments on these sections.

Round 2

Reviewer 1 Report

The quality of the language used still needs extensive edits before publication can be advised. 

Specific comments:

  1. In the title, please change "Systematic reviews and meta-analysis" to "A Systematic Review and Meta-analysis".
  2. Please change "otherwise the fixed-effects model" to "otherwise a fixed-effects model".
  3. Please change "the most overall risk of bias was low" to "for most studies, the overall risk of bias".
  4. Please change "World health organization" to "The World Health Organisation (WHO)".

Reviewer 2 Report

Thank you for addressing the previous comments.

Fig 3 appears to have an issue with the axis. Otherwise, I have no further comments on this manuscript. 
